# Characterization of Macrophages and Osteoclasts in the Osteosarcoma Tumor Microenvironment at Diagnosis: New Perspective for Osteosarcoma Treatment?

**DOI:** 10.3390/cancers13030423

**Published:** 2021-01-23

**Authors:** Anne Gomez-Brouchet, Julia Gilhodes, Nathalie Van Acker, Regis Brion, Corinne Bouvier, Pauline Assemat, Nathalie Gaspar, Sebastien Aubert, Jean-Marc Guinebretiere, Beatrice Marie, Frederique Larousserie, Natacha Entz-Werlé, Gonzague de Pinieux, Eric Mascard, Francois Gouin, Pierre Brousset, Marie-Dominique Tabone, Marta Jimenez, Marie-Cecile Le Deley, Jean-Yves Blay, Laurence Brugieres, Sophie Piperno-Neumann, Francoise Rédini

**Affiliations:** 1Department of Pathology, CHU, IUCT-Oncopole, University of Toulouse, Eq19. ONCOSARC CRCT, UMR 1037 Inserm/UT3, ERL 5294 CNRS, 31059 Toulouse, France; 2Biostatistics Unit, Institut Claudius Regaud, IUCT-Oncopole, 31059 Toulouse, France; Gilhodes.Julia@iuct-oncopole.fr; 3Department of Pathology, CHU, Imag’IN Platform, IUCT-Oncopole, 31059 Toulouse, France; vanacker.nathalie@iuct-oncopole.fr (N.V.A.); brousset.p@chu-toulouse.fr (P.B.); 4Pôle Os—Articulations—Chirurgie Plastique, University of Nantes, INSERM UMR 1238, 44035 Nantes, France; regis.brion@univ-nantes.fr (R.B.); francoise.redini@univ-nantes.fr (F.R.); 5Department of Pathology, CHU la Timone, 13005 Marseille, France; Corinne.BOUVIER2@ap-hm.fr; 6Institut de Mécanique des Fluides de Toulouse, UMR 5502 CNRS, INPT, University of Toulouse, 31400 Toulouse, France; pauline.assemat@imft.fr; 7Department of Oncology for Child and Adolescents, Gustave Roussy Cancer Center, Paris-Saclay University, 94805 Villejuif, France; Nathalie.GASPAR@gustaveroussy.fr; 8Department of Pathology, CHRU Lille, 59000 Lille, France; Sebastien.AUBERT@CHRU-LILLE.FR; 9Department of Pathology, Curie Institut, 75005 Paris, France; jean-marc.guinebretiere@curie.net; 10Department of Pathology, CHRU Nancy, 54035 Nancy, France; b.marie@chru-nancy.fr; 11Department of Pathology, AP-HP, Hôpital Cochin, Université Paris Descartes, 75014 Paris, France; frederique.larousserie@aphp.fr; 12Pediatric Oncology Department CHRU, Hautepierre, 67200 Strasbourg, France; Natacha.entz-werle@chru-strasbourg.fr; 13Department of Pathology, CHU, 37000 Tours, France; gonzague.dubouexic@univ-tours.fr; 14Department of Pediatric Orthopedic Surgery, AP-HP, Necker Hospital, 75015 Paris, France; Eric.mascard@wanadoo.fr; 15Centre Léon-Bérard Lyon, CHU Nantes, Université de Nantes, INSERM UMR 1238, 69008 Lyon, France; Francois.GOUIN@lyon.unicancer.fr; 16Pediatric Oncology Department, Armand Trousseau, APHP, 75012 Paris, France; marie-dominique.tabone@aphp.fr; 17Unicancer, Paris, France, Department of Oncology, Centre Leon-Bérard, Claude Bernard University, 69008 Lyon, France; m-jimenez@unicancer.fr (M.J.); jean-yves.blay@lyon.unicancer.fr (J.-Y.B.); 18Biostatistics Unit, Gustave Roussy Cancer Campus, 94805 Villejuif, France; m-ledeley@o-lambret.fr; 19Department of Pediatric Oncology, Gustave Roussy, 94805 Villejuif, France; Laurence.BRUGIERES@gustaveroussy.fr; 20Department of Medical Oncology, Institut Curie, 75005 Paris, France; sophie.piperno-neumann@curie.net

**Keywords:** osteosarcoma, zoledronic acid, osteoclast, macrophage, bipotent macrophage, multiplex

## Abstract

**Simple Summary:**

Due to the great genetic instability of osteosarcoma (OS), a recurrent molecular therapeutic target has not been identified to date. Therefore, characterization of the OS tumor microenvironment (TME) might offer new therapeutic perspectives. The OS2006 trial, originally designed to evaluate the impact of zoledronic acid (ZA, osteoclast-inhibitor) addition to conventional OS-therapies, was ended preliminary due to a negative impact on patient survival. Through retrospective biomarker analysis of the unique biological samples collected during the trial, we demonstrate here that ZA not only acts on harmful osteoclasts but also on protective macrophages, clarifying its detrimental effect. By multiplex immunohistochemistry, applied on additional OS biopsies, an important bipotent macrophage-population (CD168+/CD163+), homogenously distributed throughout OS tumor areas, was identified. These bipotent cells might play a determining role in the evolution of OS and offer a novel therapeutic approach. A clear definition of the macrophage populations present at diagnosis could re-enforce therapeutic decisions.

**Abstract:**

Biological and histopathological techniques identified osteoclasts and macrophages as targets of zoledronic acid (ZA), a therapeutic agent that was detrimental for patients in the French OS2006 trial. Conventional and multiplex immunohistochemistry of microenvironmental and OS cells were performed on biopsies of 124 OS2006 patients and 17 surgical (“OSNew”) biopsies respectively. CSF-1R (common osteoclast/macrophage progenitor) and TRAP (osteoclast activity) levels in serum of 108 patients were correlated to response to chemotherapy and to prognosis. TRAP levels at surgery and at the end of the protocol were significantly lower in ZA+ than ZA− patients (*p*_adj_ = 0.0011; 0.0132). For ZA+-patients, an increase in the CSF-1R level between diagnosis and surgery and a high TRAP level in the serum at biopsy were associated with a better response to chemotherapy (*p* = 0.0091; *p* = 0.0251). At diagnosis, high CD163+ was associated with good prognosis, while low TRAP activity was associated with better overall survival in ZA− patients only. Multiplex immunohistochemistry demonstrated remarkable bipotent CD68+/CD163+ macrophages, homogeneously distributed throughout OS regions, aside osteoclasts (CD68+/CD163−) mostly residing in osteolytic territories and osteoid-matrix-associated CD68−/CD163+ macrophages. We demonstrate that ZA not only acts on harmful osteoclasts but also on protective macrophages, and hypothesize that the bipotent CD68+/CD163+ macrophages might present novel therapeutic targets.

## 1. Introduction

Osteosarcoma (OS) is the most common primary malignant bone tumor [1] and has a worldwide incidence of one to three cases per million annually with a higher incidence in teenagers (0.8–1.1/100,000/year for ages 15–19). Conventional OS is the most common histologic subtype (75% of all cases) [1,2]. Microscopically, it is of high grade, highly heterogeneous, with cells that produce varying amounts of osteoid/chondroid matrix. Conventional OS are then divided in three major subtypes: osteoblastic (50%), chondroblastic (25%) and fibroblastic (25%) OS [3]. OS represents [1] radiographically mixed lesions with heterogeneous distribution of lytic and sclerotic territories [4]. Sclerosis is the result of tumoral osteoid production, and osteolysis is the result of osteoclastic resorption [4]. The high complexity of the OS genome did not allow for the identification of key molecular therapeutic targets so far [5,6,7]. Hence, OS treatment—based on chemotherapy with surgical resection—and survival rates, have not evolved over the past 4 decades. The prognosis remains poor for 25% of OS patients generally presenting metastatic relapses [8,9,10,11,12]. New therapies are needed to better control metastatic clones and increase survival. Therapeutic approaches inhibiting the PI3K-mTOR and IGF pathways, might be of interest, as shown in in vitro models [13,14], but need yet to be investigated and validated in patients with OS. Several phase II clinical trials [15,16] are currently under evaluation for new drug efficacy at diagnosis and relapse respectively. Results however are not yet available or need further investigation.

The pivotal role played by the bone Tumor Micro-Environment (TME) has been suggested for many years in OS [17,18,19]. The bone microenvironment is very complex, with a finely balanced interplay between osteoblasts, osteoclasts, chondrocytes, mesenchymal stem/progenitor cells, hematopoietic cells, endothelial cells, and others [20]. The process of bone formation by osteoblasts and bone resorption by osteoclasts is regulated by the biological activity of various factors required for bone homeostasis. An instability in this process may induce the formation of abnormal progenitors and thus result in an imbalance between the demand of progenitor cells in periods of intensive bone formation and remodeling [20]. Osteoclastic cells (OCs) have been reported to be associated with a poorer prognosis in the pathogenesis of OS [19]. Preclinical studies from our group showed that zoledronic acid (ZA)—an inhibitor of the OC function—suppressed lung metastasis and prolonged survival of OS-bearing mice [21], and that ZA in combination with a stimulator of macrophage activity enhanced primary tumor regression in pre-clinical xenogeneic and syngeneic mice models [22]. Based on the assumption of the existence of a vicious cycle between tumor cell proliferation and bone degradation by OCs [19,20,23], the French OS2006 trial [11] was developed, aiming at evaluating the impact of the addition of ZA—an inhibitor of the OC function—to conventional therapies (surgery and chemotherapy) on patient survival. The study showed that ZA did not improve the outcome and could even be detrimental on survival [11,24]. Initially, the OS2006 trial was not designed to understand the role of the TME in the biology of OS. The unique sample collection (including serum collection over time and biopsy collection at diagnosis) collected simultaneously, served to analyze the role of the TME cells in this exploratory study.

As in most sarcomas [24], the OS immune-TME mainly comprises TAM [25,26], dendritic cells, myeloid cells, OCs and different amounts of T-lymphocytes [25]. Recent studies revealed that an infiltrate rich in TAM induce immunosuppression, responsible for the inefficacy of lymphocyte-targeted therapies [24,27]. Previously, we have shown that a low amount of CD8+ T-lymphocytes, at diagnosis, appeared to be associated with increased OS metastasis in accordance with findings of Zhang and colleagues [26,28]. The beneficial role of macrophage infiltration has been repeatedly reported in OS patients [26,29,30,31]. Two main subsets of macrophages are usually taken into consideration, the pro-inflammatory M1 type and the anti-inflammatory pro-tumoral M2 type [29,32]. The polarization and infiltration of these two macrophage subtypes dynamically change [32], depending not only on the cytokines secreted by the tumor cells and/or the microenvironment’s immune cells but also on numerous other factors including genetic drivers of disease, host specific considerations (lifestyle, age, menopause status), as well as immune-related basis for disease [33,34]. Analysis of the transcriptome indicate that TAMs have a mixed phenotype expressing both M1 and M2 markers [35]. In general, TAM from established tumors have properties of M2-activated cells, enhancing epithelial-mesenchymal transition (EMT), helping cancer progression and metastasis [34,36]. In an in vitro osteosarcoma study, Zhou and colleagues demonstrated that M2 polarization of TAMs promoted pulmonary metastasis and that the STAT3/6 family and the nuclear receptor PPAR-γ were the transcription factors that translate signals and polarize macrophages into the M2 phenotype [37]. 

Despite the continuous evolution with regards to the in situ identification of M1- and M2-macrophages [32,38], no clear consensus is defined for the use of tissue-markers that formally differentiate M1 macrophages from M2. In some studies, CD68 is considered to be a pan-macrophage marker, in others it is stated to be an M1-polarized macrophage marker. In the light of bone tissue and OS, CD68 can be considered as a marker of macrophages and of OCs [39], a type of cells that plays a crucial role in bone remodeling [40]. On the other hand, CD163 is widely used as a macrophage marker [41], enabling the study of the so-called M2-type macrophages. In tumoral bone tissue, different types of resident macrophages can be found, with or without osteoclastic activity [42,43]. It is known that the monocyte/macrophage lineage gives rise to OC by the action of macrophage colony stimulating factor (M-CSF or CSF-1) and receptor activator of nuclear factor-kB ligand (RANKL) [44]. Recently, Xiao Y et al. [45] showed that monocytes/macrophages, OCs, and dendritic cells shared a common progenitor that gives rise to a bipotent macrophage/OC progenitor. Previously, we demonstrated that only CD163+ TAM infiltration was associated with a better prognosis in OS [26], and that CD68+ macrophages could oppose this positive effect on OS patient prognosis [26,29,30,31], in contrast to what happens in soft tissue tumors. Both in vivo and in vitro co-culture studies from our group and others respectively have shown that the OCs and macrophages play important roles in the development and progression of OS and metastases [22,46]. Taken together, these data confirm the relevance of studying CD68+ OCs and CD163+ macrophages in the TME of OS rather than studying M1 and M2 types of macrophages. Thus, we hypothesize that not only CD68+ OCs, but also CD163+ macrophages and even their progenitors play a crucial role in patient prognosis and that they could all represent a ZA target, and thus switching the ZA expected benefits into deleterious effects.

In order to clarify the respective roles of OCs and macrophages in the complex TME of OS and their involvement in the detrimental effect of zoledronic acid (ZA) treatment observed in the OS2006 trial [11], we combined two strategies: (1) Conventional and multiplex fluorescence immunohistochemistry were performed on biopsies of 124 OS2006 patients (included in a Tissue Micro Array, TMA) and 17 surgical “OSNew” biopsies respectively. The multiplex immunofluorescence allowed us to study the spatial distribution of CD68+ OCs, CD163+ macrophages, cytotoxic CD8+ T-cells and SATB2+ osteosarcoma cells by digital analysis; (2) Serum biomarker analysis was performed for 108 patients at diagnosis and all along the treatment of the OS2006 trial [11] and was correlated to the response of chemotherapy and to prognosis. The tartrate-resistant acid phosphatase (TRAP) activity allowed us to study OC activity [47,48]—related to the CD68+ OC biomarker—[39] and colony-stimulating factor 1 receptor (CSF-1R) enabled us to study progenitors of both CD68+ OCs and CD163+ macrophages [45].

Three important results stand out from our study: (1) The multiplex immunofluorescence digital analysis allowed us to identify a double CD68+/CD163+ positive TAM population, with unknown polarization and probably high plasticity in response to the complex interactions with the other cells of the TME, OS cells and which most likely has an impact on patient response to treatment; (2) CD68−/CD163+ macrophages, associated to a good prognosis, are primarily observed in osteoid-rich tumor regions, the predominant morphological OS type, and exert opposite effects to CD68+/CD163− OCs found predominantly in osteolytic regions; (3) ZA inhibits not only OCs but more importantly CD163+ macrophages initially associated to a good prognosis, explaining the detrimental effect of ZA in the OS2006 trial. Our data highlight the importance of studying the balance between OCs and macrophages in the development of OS and in treatment efficacy. A clear definition of the macrophage populations present at diagnosis could re-enforce therapeutic decisions.

## 2. Results

### 2.1. Description of the OS2006 Patient Population

Patient and tumor characteristics, treatment and outcomes are described in Appendix A for the entire initial OS2006 cohort with available tissue microarrays (*n* = 124) and in Table 1 for the 108 (out of 124) patients for whom additional serum assays were performed and thus included in this study. Thirty-nine of them (36.1%) received ZA (ZA+) and the remaining 69 patients received chemotherapy only (ZA−; randomized or not). We investigated the representability of the included population. No statistical difference was observed between the excluded and included patients in the present study in terms of clinical parameters. Only the chemotherapy response was better in the ZA+ subgroup whereas the overall survival remained the same.

### 2.2. Immunohistochemical Analyses of Macrophage and Osteoclast Cell Populations in OS2006 Cohort

Classic immunohistochemistry of cytotoxic T-lymphocytes (CD8+) and macrophage/osteoclastic populations (CD163+/CD68+) revealed low presence of CD8+ cells (median 1%) and predominant presence of CD163+ macrophages (median 40%) over CD68+ OCs (median 20%) (Table 2). No significant difference was noticed for these 3 markers in the studied population (ZA+, ZA−).

### 2.3. Fluorescence Multiplex Digital Analysis of the Spatial Heterogeneity of Tumor Microenvironmental Cells in “OSNew” Cohort

The TME cell population (CD8, CD163 and/or CD68) of the “OSNew” surgical biopsies (*n* = 17) was analyzed in 287 tumor regions (total of 1260.3 mm^2^), being morphologically classified into osteolytic (OL, 83.6%, 1118.3 mm^2^) and osteoid matrix rich (OM, 16.2%, 142.0 mm^2^) regions. Five biopsies did not contain OL tumor regions (29.4%), 8 biopsies did not contain OM tumor regions (47.1%). The discrepancy between the number of OM and OL regions is related to the surgical biopsy procedure applied for OS patients, which aims at including cellular rich areas (OL).

SATB2 (Figure 1, green) nuclear density, visualizing osteoblastic cells, differentiated areas showing OM (mean 427.1/mm^2^) and OL (mean 867.0/mm^2^) morphological characteristics. OM regions are characterized by an abundant bone matrix and low cellularity, whereas OL areas show high cellular density with nearly absent matrix (Figure 1A and Figure 1B, respectively) and often cystic formations.

The automated cell count revealed that CD8+ T-cells are scarce and account for 0.59% (mean) of the overall studied cell population, thus confirming the findings of the IHC analysis. CD8+ cells are however mainly present in osteolytic regions (OL: mean 0.70%; OM: mean 0.23%). CD68+ (Figure 1, purple) and CD163+ (Figure 1, red) cells seem to be differentially distributed in OM and OL regions. In OM regions (Figure 1A), the TME cell population is predominated by the macrophages of the CD68−/CD163+ subtype (mean 12.92%). OCs (CD68+/CD163−) (mean 4.29%) and double positive CD68+/CD163+ macrophages (mean 5.48%) are in minority (Figure 1, A1-A2-A3). In OL regions, however ((Figure 1B), a macrophage phenotype switch occurs resulting in a homogenous population consisting of CD68−/CD163+ macrophages (mean 10.28%) along with an important OC (CD68+/CD163−) (mean 7.46%) and double positive (CD68+/CD163+) macrophage (mean 9.26%) recruitment (Figure 1, B1-B2-B3).

### 2.4. Temporal Serum Assay Analysis of Macrophage and Osteoclast Activities (OS2006 Cohort)

A two-way repeated measure ANOVA was run on the 108 patients to determine if there were differences in TRAP activity and CSF-1R concentration over time and treatment. The results showed statistically significant differences in mean TRAP and CSF-1R over time (*p* < 0.005). The interaction term between treatment and time was significantly associated with TRAP only (*p* < 0.005). The median serum level of CSF-1R and TRAP at diagnosis was respectively 619.5 ng/mL and 9.7 U/L, with no significant difference between ZA+ or ZA− patients (Table 2, Figure 2). TRAP levels decreased during treatment in the entire selected cohort. Additionally, TRAP levels were significantly lower in the group of ZA+ patients compared to the group of ZA− patients at surgery (*p*_adj_ = 0.0011) and at the end of the protocol (*p*_adj_ = 0.0132). On the contrary, CSF-1R levels increased during treatment with no significant difference between ZA− and ZA+ patients (Table 2, Figure 2).

### 2.5. Biomarkers (Immunohistochemical and Serum Assays Analyses) Associated with Response to Chemotherapy in the ZA+ versus ZA− Patient Groups (OS2006 Cohort)

No statistical association was found between biomarkers and response to chemotherapy in the overall population (Figure 3). Exploratory subgroup analyses showed that in the group of patients treated with chemotherapy only, CD163+ staining seemed associated with a good response to chemotherapy (*p* = 0.0148) (Figure 3A, middle panel). On the other hand, in the ZA+ subgroup, increase of CSF-1R (Figure 3B) between diagnosis and surgery and a high TRAP (Figure 3C) level in serum at biopsy were both associated with a better response to chemotherapy (*p* = 0.0091 and *p* = 0.0251 respectively). The decrease of CSF-1R (Figure 3B, right panel) between diagnosis and surgery tends to be associated to a poorer response. Nevertheless, these findings were not significant after adjusting for multiple testing.

### 2.6. Clinical Parameters and Biomarkers Associated with Survival in Patients and Subgroup Analyses

After a median follow-up of 63 months (95%CI [58; 67]), thirty-four patients (31.5%) had died. The 5-year overall survival rate was estimated at 72.3% (95%CI [62.0–80.3]). Metastatic progression-free survival (MPFS) occurred in 37.0% of patients (40/108) and the five-year MPFS rate was estimated to 61.2% (95%CI [51.6; 69.5]). Univariable analyses for Overall Survival and MPFS are presented in Table 3. In the overall population, CD163 was the only biomarker significantly associated with better Overall Survival and MPFS (*p* = 0.011 and *p* = 0.019 respectively). Multivariable analyses presented in Appendix A show that these results do not remain significant after adjusting on prognostic clinical factors (histological subtype, presence of metastases at diagnosis and chemotherapy response). 

In exploratory subgroup analysis (univariable analysis, Table 3), we have observed that a high level of CD163 and a low rate of TRAP at diagnosis were associated with a better overall survival among ZA- patients (*p* = 0.026 and *p* = 0.017 respectively) and a trend was observed for metastatic progression-free survival (MPFS) (*p* = 0.081 and *p* = 0.051). In the group of patients treated with ZA (ZA+), no statistical correlation with Overall Survival or MPFS was noticed, nevertheless the magnitude of CD163 differences in prognosis was similar between the two treatment arms. For illustrative purposes, we present in Figure 4 the Kaplan Meier curves for Overall Survival and MPFS according to CD163 and TRAP levels with an arbitrary cut-off at the median.

## 3. Discussion

The goal of the present study was to try to decipher the spatial distribution of OC and macrophage cell populations in the complex microenvironment of OS and to identify the targets of Zoledronic Acid (ZA) observed to be harmful for patients during the French OS2006 clinical trial.

To better understand the role of Tissue Associated Macrophages in OS, we developed a multiplex multi-target digital analysis of the TME (CD68, CD163, CD8) in relation to bone-formation (SATB2) in “OSNew” biopsies, a strategy that has never been reported before. With this approach, we demonstrated that CD68+ and CD163+ cells appeared differentially distributed throughout morphologically distinct OS tumor regions—cellular rich osteolytic (OL) tumor regions and osteoid-matrix (OM) forming tumor regions poor in cellularity—supporting the hypothesis that these two cell types might exert different biological activities during OS development. Multiplex immunohistochemistry allowed us to identify three macrophage-like phenotypes: CD68−/CD163+, CD68+/CD163− and a double positive CD68+/CD163+ TAM. For the first time, we clearly show that CD68−/CD163+ macrophage dominate in OM tumor regions, over other macrophage phenotypes. In OL regions, we found a homogenous representation of these TAMs along with double labelled CD68+/CD163+ macrophages and CD68+/CD163− OCs. We issued the hypothesis that both CD68+/CD163− and CD68−/CD163+ macrophage subtypes might have, in the context of OS, opposite functions, and that the double CD68+/CD163+ subgroup might represent a group of bipotent macrophages with high plasticity and undefined polarization, waiting to switch to the CD68+/CD163− subtype or polarize to a CD68−/CD163+ macrophage variant in response to the different OS and TME stimuli [20,45]. When looking at the global macrophage cell population counted in the OM and OL regions of “OSNew” biopsies, an equivalent number of CD68−/CD163+, CD68+/CD163− and CD163+/CD68+ macrophage subtypes was observed. The discrepancy found with soft tissue sarcomas [24], in which CD163+ macrophages dominate over CD68+ macrophages, could be explained by the fact that CD68 is not only a macrophage-marker, but also, in the context of bone tissue, a marker of osteoclastic cells. Therefore, OCs increase the observed value of the CD68-marker in osteosarcoma. Another bias might come from the fact that biopsies are preferentially performed in OL regions (with higher cellularity), reducing the OM tumor region analyzed in this study (16%). In the same way, the micro-biopsies for the TMA used for conventional IHC were collected in these OL cellular OS regions. 

In clinical practice, the determination of the M1/M2 ratio appears a higher biologically relevant indicator to cancer prognosis compared to general TAM cell counts, with a high M1/M2 ratio relating to a better outcome for the patient [38]. However, considerable controversy exists over the in situ visualization of M1 and M2 macrophages. No clear consensus has been reached leading to inter-study disagreement. In addition, in the context of bone, the TME is even more complex to study due to the additional involvement of bone remodeling cells. The CD68+ cells stated to be M1-polarized macrophages in soft tissues, also represent in bone and OS, the osteoclastic cells [39] of the bone remodeling [40]. An instability in this process can lead to tumor development depending on the bone microenvironmental stimuli and M1/M2 distinction is even more complex to untangle [20]. That’s why we focused our study on the CD68+/CD163−, CD68−/CD163+, and CD68+/CD163+ cells to highlight the interaction between those cell populations, and their role in bone formation or resorption and effect of treatment. New multiplex immunohistochemical strategies could be developed to improve the ability to study macrophage polarization in relation to OCs, and their ratio in OS development and progression. This approach could become indispensable at the individual level to better define therapeutic strategies [30].

To enhance our understanding of the deleterious effect of ZA treatment on patient prognosis in the OS2006 trial, serum biomarker analysis was performed at diagnosis, before surgery +/− after 4 injections of ZA, after chemotherapy, one year after the end of treatment. Two key biomarkers were analyzed: CSF-1R as a biomarker for common progenitors of OCs and macrophages, and TRAP as an OC-activity biomarker. One biopsy was taken from each patient at diagnosis for subsequent conventional immunohistochemical studies of the TME cells (CD68, CD163, CD8). All data were correlated to the response to chemotherapy and overall and metastatic progression free survival. We showed that TRAP levels decrease significantly during ZA treatment (after pre-operative treatment and at the end of the treatment) as compared to the values obtained at diagnosis and one year after the end of treatment. This result is in accordance with the diminution of OC activity under zometa^®^ treatment, validating for the first time in a trial the biological activity of ZA on OC activity in OS patients. We also demonstrate the reversible effects of ZA on bone degradation after stopping its administration, confirming the preclinical data previously published [49,50]. The negative regulation of OS progression by OCs, reported by Akiyama et al. [51] and Ory et al. [21] in murine models, is confirmed as well for patients randomized in the ZA group. A high TRAP serum level at diagnosis is significantly associated with a better response to chemotherapy in ZA+ patients (and not in ZA− patients), suggesting that by decreasing OC function, patients defense to OS tumor evolvement significantly ameliorates when receiving chemotherapy in combination with ZA. This result is reinforced by the fact that a low rate of TRAP at diagnosis associates to a better overall survival of ZA− patients.

Immunohistochemical analysis of the biopsies of the OS2006 trial at diagnosis revealed that the presence of a high CD163+ macrophage proportion may be related to a better response to chemotherapy for patients who did not receive ZA and thus resulting in a better prognosis for the patient (Overall Survival and MPFS) [26]. In the group of ZA+ patients, the association between CD163 staining and prognosis was no longer observed. This might be related to a lack on statistical power of the studied group, nevertheless this supports our hypothesis that should be verified in a larger cohort. In the ZA+ subgroup, we also showed that an increase of CSF-1R between diagnosis and surgery was associated with a better response to chemotherapy, and conversely, a decrease of CSF-1R between diagnosis and surgery tends to be associated to a poorer response. We suggest thus that, in poor responder patients, ZA acts on CD163+ macrophages, cancelling their reported protective effect [26,29,30]. ZA might also act on the CD68+/CD163+ bipotent cells, by inducing deleterious polarization, an interesting hypothesis that should be verified in preclinical co-cultures [14,46], and ideally verified in a prospective clinical study. In good responder patients however, the increase of CSF-1R levels could indicate a favorable activation of macrophage progenitors and thus anti-tumoral activity. These findings reinforce our hypothesis that not only OCs, but also macrophages represent a target for bisphosphonates in OS, as already suggested for breast cancer by Junankanar et al. [52], maybe through ZA effect on bipotent cells.

From a therapeutic point of view, our finding that CD68-/CD163+ macrophages—associated to a better prognosis and better response to chemotherapy—are targeted by zoledronic acid (ZA) clarifies the bad results obtained in the OS2006 trial [11]. In addition, their predominant presence in osteoid-matrix forming tumor regions—the most frequent type of OS—indicates that administration of ZA is not the best therapeutic strategy. Even if the therapeutic value of a bisphosphonate or anti-RANKL antibody treatment is limited in OS, these drugs have proven their value in the treatment of osteolytic metastasis [17,51] and could be further proposed to the rare cases presenting highly osteolytic OS, characterized by high TRAP level in serum at diagnosis. Levels of the circulating RANKL marker, considered to be the most accurate marker for the evaluation of metastatic bone response [53], could then be useful in assessing the efficacy of such treatments in OS. The predominant role of macrophages and OCs in the OS tumor microenvironment along with the low presence of cytotoxic T-lymphocytes reinforce the idea that an immunosuppressive environment is created [54]. This opens the possibility to introduce immunomodulation through macrophages as potential therapeutic targets [12,15,22,24] or to use macrophage-related immune-checkpoint inhibitors. Other combined therapies acting on the mTOR [5,13] or STAT3 [34,37] pathways, in combination with macrophage immunomodulators could be considered to abolish acquired resistance and increase the effectiveness of therapies.

## 4. Materials and Methods 

### 4.1. OS 2006 Patients and Tumor Characteristics

All the patients with localized/metastatic high-grade osteosarcoma were prospectively enrolled in a national OS2006 study (NCT00470223) that included a randomized phase-3 trial [11]. They all had biopsies at diagnosis, then 12–13 weeks of pre-chemotherapy, followed by an excision of the primary tumor. Post-operative chemotherapy was adapted to the histological response and risk factors. Patients included in the trial were randomized to receive (ZA+) or not (ZA−) zoledronic acid (4 injections before surgery and 6 after), in association with chemotherapy. 

Biological studies (immunohistochemistry and serum assays) were conducted in parallel with the therapeutic protocol approved for the OS2006 trial. Specific informed consent for tumor samples was obtained from the patients or their parents or guardians if patients were under the age of 18 upon enrolment as described in the OS2006 protocol [11].

### 4.2. OS 2006 Tissue Microarray 

Tissue microarrays (TMA) were prepared as previously described [26] and used for immunohistochemical biomarker analysis (CD8, CD163, CD68). The TMAs were constructed with 124 patient samples and were stored at the certified NF 96–900 cancer biobank of Toulouse (BB-0033-00014) where the immunohistochemistry study was conducted. In accordance with French law, the biobank cancer collection was declared to the Ministry of Higher Education and Research (DC-2008-463) and a transfer agreement was obtained (AC-2013-1955) after approval by the ethics committee. All patient records and information were anonymized and de-identified prior to analysis. 

For the 108/124 OS2006 patients, the CSF-1R and TRAP biomarkers analysis was additionally performed on serum samples collected at four different time points: diagnosis, surgery, end of treatment and one year after the end of the treatment. 

### 4.3. ‘OSNew’ Sample Collection (Toulouse Cohort)

To complete our exploration we used 17 “OSNew” surgical biopsies coming from the OS Toulouse collection, also stored at the certified NF 96–900 cancer biobank of Toulouse (BB-0033-00014). We used these new samples to get a better representation of the heterogeneity of OS tumors, because fragments obtained were larger than after core biopsies and larger than tissue included in TMAs. Informed consent was obtained from all patients and the use of the biological specimens was approved by the local institutional review board. 

### 4.4. Immunohistochemistry

Automated classical immunohistochemical (IHC) stains were performed using the Discovery ULTRA (Roche, Ventana Medical Systems, Innovation Park Drive, Tucson, AZ, USA). After dewaxing, tissue slides were heat pre-treated using a CC1 (pH8) buffer (05424569001, Roche) for CD163 (64 min) and CD8 (32 min) IHC. Enzymatic pre-treatment was required for CD68 IHC using Protease 1 for 4 min at 37 °C (05266688001, Roche). The slides were blocked for endogenous peroxidase activity using the CM inhibitor (32 min at 37 °C) (Roche). The primary ready-to use CD68 (clone PGM1, Agilent Technologies, Santa Clara, CA, USA), CD163 (clone MRQ-26, Roche), CD8 (clone SP57, Roche), were incubated for 20 min, 32 min and 20 min respectively, at 36 °C. Targets were then linked using the OmniMap anti-rabbit (05269679001, Roche, Tucson, AZ, USA) and OmniMap anti-mouse (05269652001, Roche) HRP conjugated secondary antibodies. Visualization of the various targets was finally established using the ChromoMap DAB detection kit (05266645001, Roche). The tissue slides were counterstained using hematoxylin (05277965001, Roche) for 8 min followed by post-coloration using the Bluing reagent for 4 min at room temperature (05266769001, Roche). The slides were then dehydrated (ethanol and xylene) and mounted using xylene-based mounting.

Immunoreactivity was considered positive if detected in >1% of cells per core of 1mm, irrespective of staining intensity for CD68 (pre-osteoclastic and mature osteoclastic cells -multinucleated cells-), CD163 (macrophages), CD8 (cytotoxic T-lymphocytes). Tonsils and lymphoid nodes were used as positive controls for all the tested antibodies. 

### 4.5. Immunofluorescence Multiplex Staining

To further explore the tumor microenvironment (TME), we visualized CD8+ T-cells, CD163+ macrophages, and the CD68+ osteoclastic cells in relation to SATB2+ osteoblasts by multiplex immunofluorescence, in 17 surgical “OSNew” biopsies at diagnosis. The Discovery ULTRA (Roche, Ventana Medical Systems, Innovation Park Drive Tucson, Arizona 85755 USA) was used to automate the staining procedure. After dewaxing, the tissue slides were heat pre-treated using a CC1 (pH8) buffer (05424569001, Roche). The slides were then stained for multiplex immunofluorescence using the RUO Discovery Universal procedure (v0.00.0370) in a 4-step protocol with sequential denaturation (CC2 buffer (pH6), at 100 °C, 05279798001, Roche) after each step. The fluorochrome sequence recommendations were respected [55]. Tissue slides were subsequently incubated using the primary antibodies CD8 (clone C8/144B, M7103, Agilent technologies, Santa Clara, CA, USA) at a 1/200 dilution (in Envision Flex diluent (K800621-2, Agilent Technologies, Santa Clara, CA, USA)), CD163 (clone MRQ-26, 05973929001, Roche), CD68 (clone KP-1, 05278252001, Roche) and SATB2 (clone SATBA4B10, MSK101.05, Diagomics, Blagnac, France) at a dilution of 1/50 (in Envision Flex diluent (K800621-2, Agilent Technologies)). Targets were then linked using the OmniMap anti-rabbit (05269679001, Roche) and OmniMap anti-mouse (05269652001, Roche) HRP conjugated secondary antibodies. Visualization of the different targets was finally established using the Rhodamin6G (07988168001, Roche), RED610 (07988176001, Roche), Cy5 (07551215001, Roche) and FAM (07988150001, Roche) detection kits. The tissue slides were counterstained using Hoechst 33342 (H21492 (Thermofisher, Waltham, MA, USA), 1/500 in Discovery Diluent (Roche)) and mounted with gelatin mounting medium (GG1, Sigma-Aldrich, St Louis, MO, USA).

### 4.6. Tumor Annotation and Digital Image Analysis

Fluorescent stained whole tissue slides of “OSNew” surgical biopsies (*n* = 17), were scanned in 16 bits using the Zeiss AxioScan.Z1 (Carl Zeiss, Oberkochen, Germany) whole-slide scanner equipped with a Colibri 7 solid-state light source and appropriate filter cubes. To improve our understanding of the OS immune landscape, we quantified tumor-associated macrophages and CD8+ T-cells by multiplex immunohistochemistry and digital image analysis in order to reduce subjective/human error. In the context of osteosarcoma, due to the heterogeneity of pre-analytic treatment of the samples, we were obliged to study each case for differential pre-analytic staining variations and to perform an adaptive treatment of the output-image for correct threshold setting. Pre-analytic staining variations comprise variable affinity of the Hoechst-counterstain to nuclei, variable background staining of bone matrix, variable intensity of cell-markers. Areas presenting artefacts such as tissue folds, degraded tissue fragments, were excluded from the analysis. Initially 20 surgical biopsies were included in the study, 3 of them were excluded for absence of clear nuclear staining most likely due to an extended decalcification during tissue processing.

Using the HighPlex FL module of the HALO^®^ imaging analysis software (Indica Labs, Albuquerque, NM, USA), we studied the OS TME in 2 morphologically distinct tumor regions annotated by a certified pathologist (AGB). These tumor regions were selected based on their osteolytic (OL) and osteoid matrix-rich (OM) morphological characteristics. In total, 240 different OL (83.6%) and 47 OM (16.4%) regions of interest were selected. Cells stained with a cytoplasmic/membranous intensity exceeding the settings threshold were counted as positive for the full range of staining intensity. Due to the differences in bone matrix content of the 2 tumor regions, results were represented as the percentage of the cells positive for the defined phenotypes; SATB2+, CD8+, CD163+, CD68+ and the CD68+/CD163+, CD68−/CD163+, CD68+/CD163− populations. The cellular density for these different phenotypes (Number per mm2) were also recorded.

### 4.7. CSF-1R and TRAP Assays in Patient Serum 

The biomarkers (CSF-1R and TRAP) were analyzed in serum at the four previously mentioned timepoints (*n* = 108). The serum was processed, aliquoted and stored at −80 °C for further analysis. CSF-1R was assayed in 50 µL (diluted at 1/50) by the Bio-Plex^®^ system (Luminex, Bio-Rad, Marnes-la-Coquette, France) using the human magnetic Luminex^®^ assays M-CSFR/CD115 (R&D Systems Europe, Lille, France). MicroVue Bone TRAP5b ELISA (Quidel, San Diego, CA, USA) was used for TRAP activity analysis in 50 µL of serum. For both assays, the procedure was performed according to the manufacturer’s protocols. Each target concentration was calculated using, respectively, a 5-parameter and 4-parameter logistic fit curve generated from the standards. 

### 4.8. Statistical Analysis

The data was summarized by frequency and percentage for the categorical variables and by median and range for the continuous variable. Links with histological response were assessed with the Mann-Whitney U test. Overall Survival was defined as the time from inclusion to death from any cause (event) or the last follow-up (censored data). Metastatic progression-free survival (MPFS) was defined as the time from inclusion to metastatic progression or death (event) or the last follow-up (censored data). Patients presenting local relapse as first event were censored at this date.

All the survival rates were estimated by the Kaplan-Meier method, with 95% confidence intervals (CI). Univariable and multivariable analyses, adjusting the effect of biomarkers on clinical prognostic factors, were performed using the Cox proportional hazards model. Univariable analyses have been carried out in each treatment arm (ZA− and ZA+). Due to the small number of patients and the exploratory nature of this analysis, no multiple testing correction was applied. A two-way repeated measure ANOVA was run in order to determine if there were differences in TRAP and CSF-1R concentrations over time and treatment. Bonferroni’s adjustments were made to account for multiple testing. Two-sided *p*-values < 0.05 were considered statistically significant. All statistical analyses were performed using STATA 16.0 software.

## 5. Conclusions

The results of the study show that not only OCs but also macrophages represent a target for ZA, explaining its negative impact on OS2006 patient prognosis [11]. ZA treatment might not be appropriate for OS patients for whom osteoid matrix formation is predominant. The multiplex immunohistochemical study of the OS TME and clear definition of the macrophage populations present at diagnosis could re-enforce therapeutic decisions. The fact that we found double stained CD68+/CD163+ macrophages, true bipotent cells with undefined polarization, stresses the importance of the balance between macrophages and OCs in OS development, progression (Figure 5A) and response to treatment (Figure 5B). Those bipotent cells might play a significant role in the protective/harmful immune cell balance and thus on the patient response to treatment (proposed model Figure 5) and could be of interest as therapeutic target in the future. Studying the bipotent macrophage cell population in culture models and their polarization under various treatments could help to identify new therapeutic strategies. Our data highlight the importance to examine the OS tumor microenvironment in an attempt to propose the most appropriate treatment for OS patients.

## Figures and Tables

**Figure 1 cancers-13-00423-f001:**
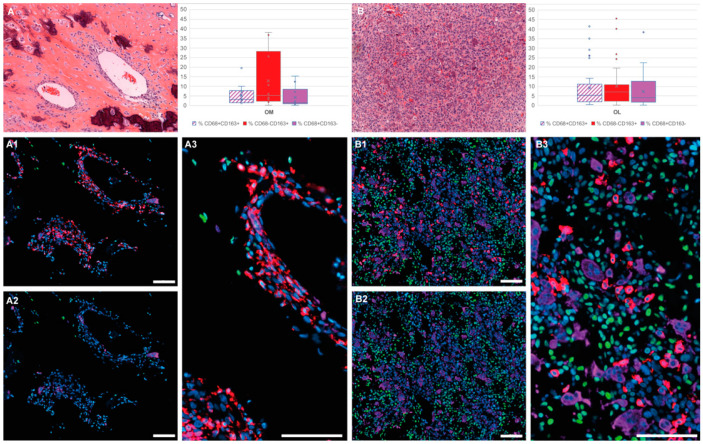
Macrophage and osteoclast population studied by multiplex immunohistochemistry. Histological and immunofluorescent representation of osteoid matrix forming (A, OM), osteolytic (B, OL) tumor regions in OS. (**A**) and (**B**) Representative region demonstrated with Hematoxylin and eosin staining (H&E). The histograms represent the percentage of the different cell populations in OM (left) and OL (right) regions: CD68+CD163+ (dashed white/red), CD68−CD163+ (red), CD68+CD163− (purple). (**A**) Histological and immunofluorescent (**A1**–**A3**) representation of cell populations present in OM tumor regions of OS. These regions are characterized by low SATB2-density (**A1**–**A3**, green) and a predominant CD163+ macrophage population (**A1**–**A3**, red) (CD68+ osteoclasts: purple). (**B**) Histological and immunofluorescent (**B1**–**B3**) representation of cell populations present in OL tumor regions of OS. These regions are characterized by high SATB2-density (**B1**–**B3**, green) and a predominant CD163+ macrophage population (**B1**–**B3**, red) and an important osteoclast-infiltrate (**B1**–**B3**, purple). Hoechst or hematoxylin nuclear counterstain: Blue. Scale bar A1-A2-A3, B1-B2-B3: 100 µm.

**Figure 2 cancers-13-00423-f002:**
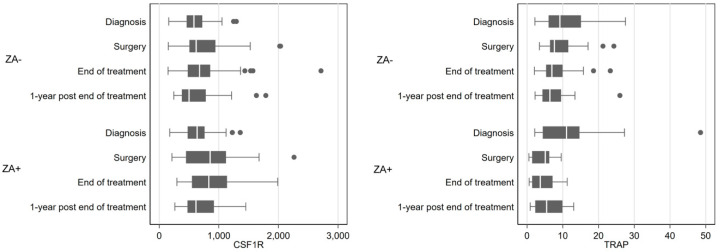
CSF-1R and TRAP serum levels over time for patients who received (ZA+) or not (ZA−) zoledronic acid.

**Figure 3 cancers-13-00423-f003:**
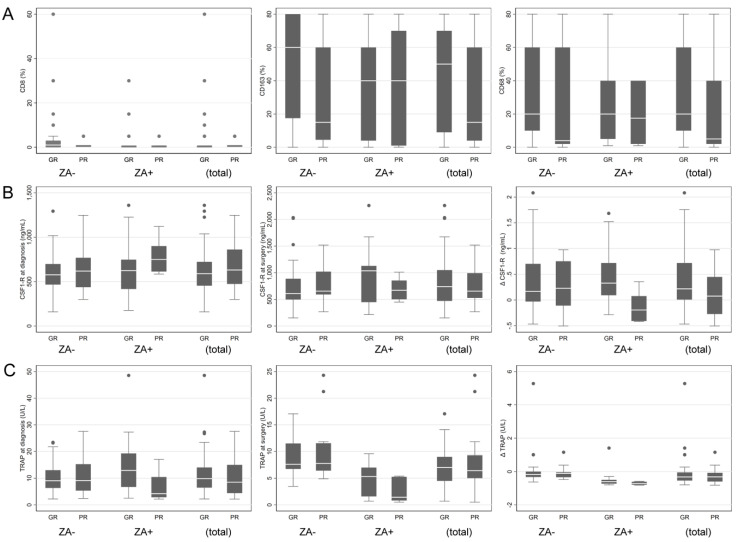
Correlation of biomarkers with response to chemotherapy for patients who received (ZA+) or not (ZA−) zoledronic acid. (**A**) Representation of response to chemotherapy (GR: good responders; PR: poor responders) in relation to immunohistochemical analysis (% of cells positive) of CD8 (**left panel**), CD163 (**middle panel**) and CD68 (**right panel**). (**B**) Representation of response to chemotherapy (GR: good responders; PR: poor responders) in relation to serum analysis of CSFR-1 at diagnosis (**left panel**), surgery (**middle panel**) and the difference of CSF-1R levels between diagnosis and surgery (**right panel**). (**C**) Representation of response to chemotherapy (GR: good responders; PR: poor responders) in relation to serum analysis of TRAP at diagnosis (**left panel**), surgery (**middle panel**) and the difference of TRAP levels between diagnosis and surgery (**right panel**).

**Figure 4 cancers-13-00423-f004:**
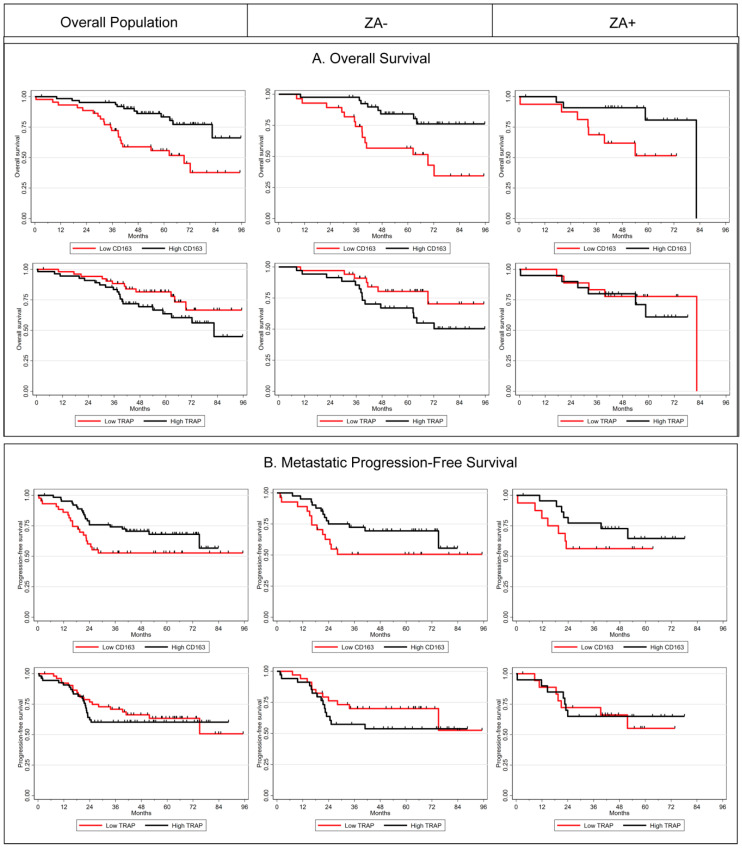
Kaplan-Meier curves according to CD163 and TRAP levels in the overall population (**left panel**), and in ZA− (**middle panel**) and ZA+ (**right panel**) groups for overall survival (**A**) and metastatic progression-free survival (**B**).

**Figure 5 cancers-13-00423-f005:**
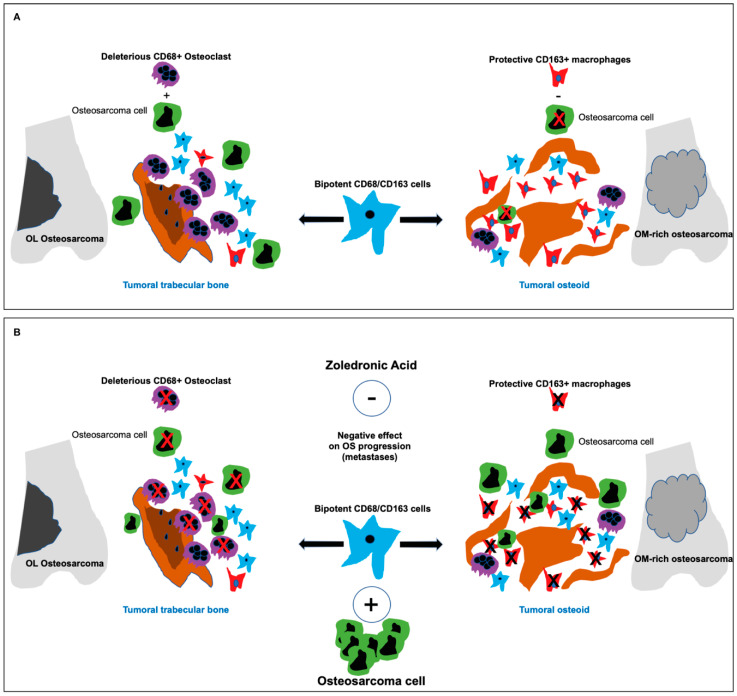
(**A**) The imbalance of osteoclasts, macrophages and bipotent cells involved in tumoral bone remodeling in Osteosarcoma. (**B**) Inhibitory effect of zoledronic acid (ZA) on both deleterious CD68+/CD163− osteoclasts and protective CD68−/CD163+ macrophages in Osteosarcoma.

**Table 1 cancers-13-00423-t001:** Demographic, clinical and histological data of the OS2006 patient cohort included in the present study (*n* = 108).

OS2006 Patient Data	Included Patients			
	Total	ZA−	ZA+	*p*
	*n* = 108	*n* = 69	*n* = 39	
**Age**				
Med (Range)	15 (5–50)	16 (5–48)	15 (8–50)	0.7907
<18y	72 (66.7%)	48 (69.6%)	24 (61.5%)	0.3954
≥18y	36 (33.3%)	21 (30.4%)	15 (38.5%)	
**Sex**				
Male	65 (60.2%)	42 (60.9%)	23 (59.0%)	0.8468
Female	43 (39.8%)	27 (39.1%)	16 (41.0%)	
**Limb vs. Axial**				
Axial	12 (11.1%)	7 (10,1%)	5 (12.8%)	0.7534
Limb	96 (88.9%)	62 (89.9%)	34 (87.2%)	
**Conventional osteosarcoma**				
Chondroblastic	28 (25.9%)	20 (29.0%)	8 (20.5%)	0.6858
Osteoblastic	68 (63.0%)	42 (60.9%)	26 (66.7%)	
Fibroblastic	6 (5.6%)	3 (4.3%)	3 (7.7%)	
Other	6 (5.6%)	4 (5.8%)	2 (5.1%)	
**Meta vs. non meta**				
Localized	86 (79.6%)	55 (79.7%)	31 (79.5%)	0.9780
Metastases	22 (20.4%)	14 (20.3%)	8 (20.5%)	
**Chemotherapy**				
API-AI	13 (12.0%)	6 (8.7%)	7 (17.9%)	0.2180
MTX	95 (88.0%)	63 (91.3%)	32 (82.1%)	
**Chemotherapy response**				
GR	67 (65.0%)	38 (57.6%)	29 (78.4%)	**0.0336**
PR	36 (35.0%)	28 (42.4%)	8 (21.6%)	
Missing	5	3	2	

**Table 2 cancers-13-00423-t002:** A: Immunohistochemical analysis of macrophage, osteoclast and T-lymphocyte markers; B. serum levels of the OS2006 patients (*n* = 108). Bonferroni’s adjustments were made to account for multiple testing.

Included Patients	Total	ZA−	ZA+	*p* _adj_
	*n* = 108	*n* = 69	*n* = 39	
**A.IHC**				
**CD8 (%)**				
Med (Range)	1 (0–60)	1 (0–60)	1 (0–30)	1.0000
Missing	12	9	3	
**CD163 (%)**				
Med (Range)	40 (0–80)	40 (0–80)	40 (0–80)	1.0000
Missing	27	18	9	
**CD68 (%)**				
Med (Range)	20 (0–80)	20 (0–80)	20 (0–80)	1.0000
Missing	12	7	5	
**B.Serum levels**				
**At diagnosis**				
**CSF-1R (ng/mL)**				
Med (Range)	619.5 (159.0–1359.5)	582.0 (159.0–1293.7)	635.3 (175.7–1359.5)	1.0000
Missing	4	2	2	
**TRAP (U/L)**				
Med (Range)	9.7 (2.2–48.5)	9.2 (2.2–27.5)	11.1 (2.2–48.5)	1.0000
Missing	3	1	2	
**At surgery**				
**CSF-1R (ng/mL)**				
Med (Range)	683.1 (150.8–2262.1)	619.2 (150.8–2034.8)	854.1 (213.6–2262.1)	1.0000
Missing	38	26	12	
**TRAP (U/L)**				
Med (Range)	6.9 (0.5–24.3)	7.6 (3.5–24.3)	5.1 (0.5–9.6)	**0.0011**
Missing	41	27	14	
**At the end of treatment**				
**CSF-1R (ng/mL)**				
Med (Range)	716.9 (146.9–2714.5)	676.3 (146.9–2714.5)	831.8 (296.1–1988.1)	1.0000
Missing	34	22	12	
**TRAP (U/L)**				
Med (Range)	6.5 (0.6–23.3)	7.0 (2.1–23.3)	3.7 (0.6–11.2)	**0.0132**
Missing	36	24	12	
**One year after the end of treatment**				
**CSF-1R (ng/mL)**				
Med (Range)	518.7 (244.1–1788.4)	499.6 (244.1–1788.4)	615.9 (261.6–1452.4)	1.0000
Missing	55	34	21	
**TRAP (U/L)**				
Med (Range)	6.2 (0.9–26.0)	6.4 (2.2–26.0)	5.5 (0.9–13.1)	1.0000
Missing	57	34	23	

**Table 3 cancers-13-00423-t003:** Univariable analyses of biomarkers associated with overall survival and metastatic progression-free survival and exploratory subgroup analyses according to the treatment (*n* = 108).

**Biomarker**	**Overall Survival**					
	All		ZA−		ZA+	
	HR [95%CI]	*p*	HR [95%CI]	*p*	HR [95%CI]	*p*
**CD8**	1 [0.7; 1.6]	0.844	1.1 [0.7; 1.7]	0.61	0 [0; 4.6]	0.077
**CD163**	0.8 [0.7; 1]	**0.011**	0.8 [0.7; 1]	**0.026**	0.8 [0.7; 1.1]	0.184
**CD68**	0.9 [0.7; 1]	0.075	0.8 [0.7; 1]	0.054	1 [0.7; 1.3]	0.894
**CSF-1R**	1 [0.9; 1.2]	0.926	1 [0.8; 1.2]	0.96	1 [0.8; 1.3]	0.927
**TRAP**	1 [1; 1.1]	0.192	1.1 [1; 1.2]	**0.017**	1 [0.9; 1.1]	0.706
	**Metastatic Progression Free Survival**					
	All		ZA−		ZA+	
	HR [95%CI]	*p*	HR [95%CI]	*p*	HR [95%CI]	*p*
**CD8**	1.1 [0.7; 1.7]	0.807	1.2 [0.7; 1.9]	0.52	0.7 [0.2; 2.7]	0.609
**CD163**	0.9 [0.8; 1]	**0.019**	0.9 [0.7; 1]	0.081	0.8 [0.7; 1.1]	0.131
**CD68**	0.9 [0.8; 1]	0.147	0.9 [0.8; 1.1]	0.183	0.9 [0.7; 1.2]	0.557
**CSF-1R**	1 [0.9; 1.2]	0.514	1.1 [0.9; 1.3]	0.305	1 [0.8; 1.2]	0.887
**TRAP**	1 [1; 1.1]	0.214	1.1 [1; 1.1]	0.051	1 [0.9; 1.1]	0.888

## Data Availability

Data is contained within the article or Appendix A.

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
