# Peer review of "Characterization of Macrophages and Osteoclasts in the Osteosarcoma Tumor Microenvironment at Diagnosis: New Perspective for Osteosarcoma Treatment?"

_cancers, 2021, doi:10.3390/cancers13030423_

Round 1

Reviewer 1 Report

Manuscript entitled "Characterization of macrophages and osteoclasts in the Osteosarcoma tumor microenvironment at diagnosis: New perspective for Osteosarcoma treatment?"

it is an observational study using staining on clinical samples. The authors identifies some correlations but not confirmed a mechanism. As the authors may know, ZA might not only altered macrophage but also other cellular response in cancer cells. Accordingly,  the authors should not make over-interpretation of the results. This work will be improved by either:

  1. Adding cell like or animal model to confirm their hypothesis or
  2. Deciphering molecular signaling aspects in tumor and macrophages

Author Response

Response to Reviewer 1 Comments

Point 1: English language and style are fine/minor spell check required

Response 1: We have revised the manuscript for spelling and found some minor errors which have been corrected. We kept track of changes made to the article in the re-submitted revised version.

Point 2: Introduction, design, results, methods and conclusion can be improved.

Response 2: We agree that the article presentation could be improved. We changed both introduction and conclusion sections in an attempt to clarify the design of this observatory and exploratory study. The pre-clinical work that has been performed by our group previously (on several animal models for osteosarcoma) and others has been added to clarify our motivation for study design and reporting.

Changes made to the text can be found at:

Lines 96-100, 109-113, 154-156

Point 3:  It is an observational study using staining on clinical samples. The authors identifies some correlations but not confirmed a mechanism.

Response 3: We agree that we do not confirm a possible mechanism but we demonstrate clearly that ZA not only acts on osteoclasts but also on the protective macrophages.

We have adjusted the article accordingly and specified that we have “identified” the protective macrophages and not only the deleterious osteoclasts as targets of zoledronic acid. A more detailed argumentation can be found in our response 4.

Changes made to the text can be found at:

Lines 64-65,158-161, 301-303

Point 4:  As the authors may know, ZA might not only altered macrophage but also other cellular response in cancer cells. Accordingly,  the authors should not make over-interpretation of the results. This work will be improved by either:

  1. Adding cell like or animal model to confirm their hypothesis or
  2. Deciphering molecular signalling aspects in tumor and macrophages

Response 4: We agree with the reviewer's comment and would like to provide some additional information and provide additional argumentation for the proposed design of this exploratory and thus observatory study.

The goal of this study was to characterize cells of the OS microenvironment to better understand their involvement in OS formation. To answer to this question we designed this exploratory study as follows : 1. We performed classical immunohistochemical stains on the OS2006 cohort and multiplex immunofluorescent stains on a new OS cohort composed of larger tissue samples to ensure optimal examination of the tumor microenvironment. 2. Additionally, we analysed blood samples collected at different time points during treatment (chemotherapy +/- ZA) in an attempt to identify the potential targets of ZA which could explain the detrimental effect observed in human subjects. Correlations were studied between the biomarkers (IHC and serum analysis) and response to treatment at one hand and metastasis-free overall survival at the other.

By applying multiplex immunohistochemistry we identified a “bipotent” macrophage cell population double stained for CD68+/CD163+ and distributed evenly in the osteolytic and osteoid-matrix rich tumor regions. We hypothesize that their role might be important in the response of the patient to treatment. Indeed, it has been reported that tumor associated macrophages (TAMs) show an important plasticity and that their polarization (M1/M2) affects response to treatment. We had to abandon the classically described M1/M2 concept for bone tissue because the markers used and described for soft tissue are not appropriate in bone. In particular, CD68 considered to be a M1-marker in soft tissue is the marker of excellence for osteoclasts known to be harmful in primary and even secondary bone tumors.

Before starting the OS2006 trial, we performed several pre-clinical studies using animal models for osteosarcoma in which we demonstrated that the combination of MEPACT® and ZA reduced the volume of the primary tumor, but that ZA did not improve the effect of the MEPACT® on metastasis occurrence. The intricate role of macrophages and osteoclasts in OS tumor biology was demonstrated in these animal models studied but appeared different to what we observed in human subjects. That’s why we decided to study in more detail the macrophage and osteoclast populations on the unique and large sample cohort of the OS2006 trial. We agree however that for identifying a true mechanism of action for ZA on these microenvironmental cell populations we need to design a new pre-clinical study in which we will investigate the “bipotent” CD68+/CD163+ cells and their polarization under the action of different therapeutic agents. The results of the Sarcome-13/OS2016 trial will certainly help us to orient the experimental design.

Changes made to the text can be found at:

Lines 96-100, 109-113, 131-139, 154-156, 158-161, 301-303, 375-377

Reviewer 2 Report

The authors of the present work explored the role of osteoclasts and macrophages in the osteosarcoma TME and their involvement in the detrimental effect of zoledronic acid observed in the OS2006 trial. They evaluate the spatial distribution of CD68+ OCs, CD163+ macrophages, cytotoxic CD8+ T-cells and SATB2+ osteosarcoma through standard and multiplex fluorescence IHC on 124 osteosarcoma patients included in OS2006 trial and 17 new biopsies. Moreover CSF-1R and TRAP levels in serum of 108 patients was assessed and correlated to the response to chemotherapy and prognosis. The results suggested the presence of a double CD68+/CD163+ positive TAM population which could be involved in patient response to treatment. Furthermore CD68-/CD163+ macrophages were primary observed in osteoid-rich tumor regions and exert opposite effects to CD68+/CD163- OCs found predominantly in osteolytic regions. Finally, zoledronic acid inhibits OCs and CD163+ macrophages, initially associated to a good prognosis, providing explanation of its detrimental effect observed in the OS2006 trial.

The manuscript is well written and organized. The authors Gomez-Brouchet et al. have explored a hot topic. 

The manuscript would benefit from the following:

  • The role of macrophages in osteosarcoma has been already investigated in some translational studies. The authors should added to the references the following works: “Tumor-associated macrophages promote lung metastasis and induce epithelial-mesenchymal transition in osteosarcoma by activating the COX-2/STAT3 axis” Cancer Lett. 2019 Jan;440-441:116-125. doi: 10.1016/j.canlet.2018.10.011. “All-Trans Retinoic Acid Prevents Osteosarcoma Metastasis by Inhibiting M2 Polarization of Tumor-Associated Macrophages” Cancer Immunol Res. 2017 Jul;5(7):547-559. doi: 10.1158/2326-6066.CIR-16-0259.
  • RANKL serum analysis should be performed and correlated to the response to chemotherapy and prognosis. The work titled “RANKL: A promising circulating marker for bone metastasis response” Oncol Lett. 2016 Oct;12(4):2970-2975. doi: 10.3892/ol.2016.4977, should be included in the manuscript.
  • In the discussion section, the authors reported that TRAP levels decrease with zoledronic acid treatment as confirmed by the diminution of osteoclasts activity under zometa treatment. This data is consistent with other works in which osteoclast activity and differentiation was abrogated with the exposure to zoledronic acid and with bone-targeted therapy in other bone lesions. The authors should include the references: “The effect of everolimus in an in vitro model of triple negative breast cancer and osteoclasts” Int J Mol Sci. 2016 Nov 1;17(11):1827.doi: 10.3390/ijms17111827 and “CSF-1 blockade impairs breast cancer osteoclastogenic potential in co-culture systems” Bone. 2014 Sep;66:214-22. doi: 10.1016/j.bone.2014.06.017. Epub 2014 Jun 20.
  • In the discussion the authors should underline the importance of the axis PI-3K/mTOR together with macrpphages as possible targets for osteosarcoma. In this regard the reference “Preclinical Effectiveness of Selective Inhibitor of IRS-1/2 NT157 in Osteosarcoma Cell Lines” Front Endocrinol (Lausanne). 2015 May 13;6:74. doi: 10.3389/fendo.2015.00074 should be included.

Major revision are requested before publication.

Author Response

Response to Reviewer 2 Comments

The authors of the present work explored the role of osteoclasts and macrophages in the osteosarcoma TME and their involvement in the detrimental effect of zoledronic acid observed in the OS2006 trial. They evaluate the spatial distribution of CD68+ OCs, CD163+ macrophages, cytotoxic CD8+ T-cells and SATB2+ osteosarcoma through standard and multiplex fluorescence IHC on 124 osteosarcoma patients included in OS2006 trial and 17 new biopsies. Moreover CSF-1R and TRAP levels in serum of 108 patients was assessed and correlated to the response to chemotherapy and prognosis. The results suggested the presence of a double CD68+/CD163+ positive TAM population which could be involved in patient response to treatment. Furthermore CD68-/CD163+ macrophages were primary observed in osteoid-rich tumor regions and exert opposite effects to CD68+/CD163- OCs found predominantly in osteolytic regions. Finally, zoledronic acid inhibits OCs and CD163+ macrophages, initially associated to a good prognosis, providing explanation of its detrimental effect observed in the OS2006 trial.

The manuscript is well written and organized. The authors Gomez-Brouchet et al. have explored a hot topic.

The manuscript would benefit from the following:

Point 1:

The role of macrophages in osteosarcoma has been already investigated in some translational studies. The authors should added to the references the following works:

“Tumor-associated macrophages promote lung metastasis and induce epithelial-mesenchymal transition in osteosarcoma by activating the COX-2/STAT3 axis” Cancer Lett. 2019 Jan;440-441:116-125. doi: 10.1016/j.canlet.2018.10.011. “All-Trans Retinoic Acid Prevents Osteosarcoma Metastasis by Inhibiting M2 Polarization of Tumor-Associated Macrophages” Cancer Immunol Res. 2017 Jul;5(7):547-559. doi: 10.1158/2326-6066.CIR-16-0259.

Response 1: We revised the manuscript and added requested references: Han et al, 2018: reference 34, Zhou et al, 2017: reference 37.

Changes to the text can be found at:

Lines 129-139, 136-139, 395-398.

Point 2:

RANKL serum analysis should be performed and correlated to the response to chemotherapy and prognosis. The work titled “RANKL: A promising circulating marker for bone metastasis response” Oncol Lett. 2016 Oct;12(4):2970-2975. doi: 10.3892/ol.2016.4977, should be included in the manuscript.

Response 2: We revised the manuscript and added requested references: Ibrahim et al, 2016: reference 53. We propose in conclusion that RANKL can be used as a marker for follow-up of the response to treatment.

Changes to the text can be found at:

Lines 389-391

Point 3:

In the discussion section, the authors reported that TRAP levels decrease with zoledronic acid treatment as confirmed by the diminution of osteoclasts activity under zometa treatment. This data is consistent with other works in which osteoclast activity and differentiation was abrogated with the exposure to zoledronic acid and with bone-targeted therapy in other bone lesions. The authors should include the references: “The effect of everolimus in an in vitro model of triple negative breast cancer and osteoclasts” Int J Mol Sci. 2016 Nov 1;17(11):1827.doi: 10.3390/ijms17111827 and “CSF-1 blockade impairs breast cancer osteoclastogenic potential in co-culture systems” Bone. 2014 Sep;66:214-22. doi: 10.1016/j.bone.2014.06.017. Epub 2014 Jun 20.

Response 3: We revised the manuscript and added requested references: Mercatali et al, 2016: reference 14, Liverani et al, 2014: reference 46.

Changes to the text can be found at:

Lines 96-100, 154-156, 375-377

Point 4:

In the discussion the authors should underline the importance of the axis PI-3K/mTOR together with macrophages as possible targets for osteosarcoma. In this regard the reference “Preclinical Effectiveness of Selective Inhibitor of IRS-1/2 NT157 in Osteosarcoma Cell Lines” Front Endocrinol (Lausanne). 2015 May 13;6:74. doi: 10.3389/fendo.2015.00074 should be included.

Response 4: We revised the manuscript and added requested references: Garofali et al, 2015: reference 13.

Changes to the text can be found at:

Lines 96-100, 395-398

Point 5: Introduction and conclusion section can be improved.

Response 5: In accordance to the reviewers suggestions and comments we revised the introduction and conclusion section in an attempt to clarify the design of this exploratory study. The pre-clinical work that has been performed by our group previously (on several animal models for osteosarcoma) has been added to clarify our motivation for study design and reporting. We kept track of changes made to the article in the re-submitted revised version.

Round 2

Reviewer 2 Report

The work has been improved. The authors have addressed some important issues raised. Now the paper could be considered for publication

The paper could be considered for the publication.